# The Interface Strengthening of Multi-Walled Carbon Nanotubes/Polylactic Acid Composites via the In-Loop Hybrid Manufacturing Method

**DOI:** 10.3390/polym15224426

**Published:** 2023-11-16

**Authors:** Hongbin Li, Zhuang Jiang, Zhihua Li, Yubao Peng, Qiushuang Zhang, Xinyi Xiao

**Affiliations:** 1Department of Electromechanical Engineering, Qingdao University of Science and Technology, Qingdao 266061, China; qklhb1989@qust.edu.cn (H.L.); jiangz335553@163.com (Z.J.); lizhihua127@163.com (Z.L.); 2023031008@mails.qust.edu.cn (Y.P.); qs.zhang@qust.edu.cn (Q.Z.); 2School of Mechanical Engineering, Tianjin University, Tianjin 300354, China; 3Department of Mechanical Engineering, University of North Texas, Discovery Park, Denton, TX 76203, USA

**Keywords:** in-loop hybrid manufacturing, multi-wall carbon nanotubes, interfacial property, molecular dynamics

## Abstract

In this study, a new in-loop hybrid manufacturing method is proposed for fabricating multi-walled carbon nanotube (MWCNTs)/polylactic acid (PLA) composites. Molecular dynamics simulations were conducted in conjunction with experiments to reveal the mechanism of the proposed method for improving the interfacial performance of MWCNTs/PLA. The superposed gradients in the PLA chain activity and conformation due to the plasma-actuating MWCNTs promoted intermolecular interaction and infiltration between the MWCNTs and PLA chains, forming an MWCNTs-stress-transfer bridge in the direction perpendicular to the interlayer interface, and finally enhancing the performance of the composites. The experimental results indicated that the interfacial shear strength of the specimen fabricated using the proposed method increased by 30.50% to 43.26 MPa compared to those without the addition of MWCNTs, and this value was 4.77 times higher than that of the traditional manufacturing method, demonstrating the effectiveness of the proposed method in improving the interfacial properties of MWCNTs/PLA composites.

## 1. Introduction

Additive manufacturing (AM) has ushered in a paradigm shift in the realm of prototyping and manufacturing. It has opened up avenues for rapid prototyping, intricate customization, and the production of lightweight components imbued with integrated structures and functionalities [1]. Among the various AM methods, Material Extrusion (MEX) stands as a prominent choice. MEX leverages a swift material deposition process, achieved through selectively dispensing polymers through a nozzle. As a type of MEX, fused filament fabrication (FFF) technology finds extensive utility in conceptual prototyping, as well as the fabrication of end-use parts [2]. The formation of objects through FFF hinges on the incomplete diffusion of interface molecules, a phenomenon intricately linked with temperature-dependent cooling. The interfaces between adjacent material beads are primarily established through physical bonding and intermolecular forces [3], which serve as the principal agents governing structural integrity under loading conditions. The quality of FFF-produced components has a significant impact on their microstructure and mechanical properties [4,5,6]. In FFF, thermoplastic polymers undergo a two-step transformation: they are initially heated and subsequently rapidly solidified as they exit the nozzle. This process, known as Fused Filament Fabrication, leads to a brief period, during which, the matrix material remains above the glass transition temperature. Consequently, there is limited adhesion between interlaminar beads when they come into contact with the newly deposited material. The resulting weak interlaminar bonding emerges as the predominant factor impeding the performance of FFF-produced parts. Enhancements in this interlaminar bonding can be achieved through a comprehensive investigation of the tensile [1,7], compression [8], torsional [9], and fatigue properties [10,11,12] of these interfaces. This, in turn, underscores the significance of optimizing FFF processing parameters and applying post-processing techniques [13] to overcome these challenges.

While considerable strides have been made in augmenting interlaminar bonding properties by optimizing FFF processing parameters [14,15,16] and implementing post-processing techniques [17,18,19,20], there still exist limitations in meeting the exacting requirements of engineering applications. 

Multi-walled carbon nanotubes (MWCNTs), renowned for their exceptional rigidity, high aspect ratio fillers, and remarkable mechanical properties [21,22,23], have emerged as a prime choice for reinforcing polymers in recent years. Numerous research endeavors have sought to fabricate nanocomposites using the Fused Filament Fabrication (FFF) process [23,24,25,26,27,28]. This entails blending MWCNTs with granular matrix materials in a specific ratio to form filament feedstock through a screw extruder. Subsequently, the filamentous feedstock is introduced into an FFF printer to craft MWCNT-reinforced composites [22,29,30]. The tensile strength of the MWCNT composites produced using this method significantly surpasses that of matrix materials devoid of MWCNTs [22,29,30,31]. Notably, the tensile strength of these MWCNT composites can reach an impressive 61.3 MPa when the MWCNT content stands at 1 wt.% [30]. Nevertheless, when the MWCNT content exceeds this threshold, as observed with concentrations greater than 1 wt.% [30], challenges surface due to the limited melt flow index of the matrix material. This low melt flow index hinders the extrusion of the composite feedstock and precipitates the formation of aggregates within the matrix. Consequently, this poses difficulties in achieving a homogenous dispersion of MWCNTs, a phenomenon critical to the composite’s overall performance. The presence of such aggregates adversely impacts the bonding between interlaminar beads [22,27,30,31,32]. As a consequence, the resulting materials become prone to vulnerability and brittleness, rendering them susceptible to fracture. This ultimately obstructs the potential enhancement of the nanocomposite’s properties [25,33,34]. In summary, while MWCNTs hold promise as formidable reinforcement materials, achieving optimal dispersion and maintaining the integrity of interlaminar bonding remains a critical consideration in the quest to unlock the full potential of these nanocomposites. Table 1 provides an overview of the current strategies for enhancing interfacial shear strength.

A substantial body of prior research has predominantly directed its focus towards elucidating the effects of nanofillers on the macroscopic properties of Multi-walled Carbon Nanotube (MWCNTs) nanocomposites [29,30,31]. These macro properties typically encompass bulk characteristics, such as tensile strength, modulus of elasticity, and thermal conductivity, among others. While these investigations have yielded valuable insights into the overall performance of these composites, there has been a notable paucity of attention directed toward comprehensively examining the influence of nanofillers on the interfacial properties situated between interlaminar beads [25]. Interlaminar beads refer to the boundaries or interfaces between the adjacent layers or beads within a composite structure, typically formed during the FFF process. These interfacial properties encompass a range of attributes, including the interlaminar adhesion, bonding strength, and compatibility between the nanofillers and the surrounding matrix material. Such properties are of paramount importance in determining the composite’s structural integrity, resistance to delamination, and overall mechanical behavior. Despite the crucial role that interlaminar properties play in shaping the performance of nanocomposites, research has somewhat overlooked their detailed examination. Consequently, there remains a significant gap in our understanding of how nanofillers, like MWCNTs, impact these interfacial properties. A more comprehensive exploration of this aspect is critical, as it can unveil strategies for optimizing the design and manufacturing of nanocomposites for improved interlaminar performance and, by extension, enhance the material’s overall mechanical and functional attributes.

This is partly due to the inherent characteristics of Fused Filament Fabrication (FFF), wherein most nanofillers tend to align themselves predominantly along the direction of the filament beads and do not effectively infiltrate adjacent layers. Consequently, MWCNTs’ potential in enhancing interfacial properties has not been fully realized.

This paper introduces a novel in-loop hybrid manufacturing (HM) approach, which integrates plasma dispersion and FFF technologies. In this method, MWCNTs are promptly sprayed onto the surface of Polylactic Acid (PLA) beads as the matrix material is extruded from the FFF printer during an in-loop conversion process. The research aims to explore the impact of MWCNT addition on the interfacial properties of the adjacent matrix layers. This exploration is conducted through a combination of molecular dynamics simulations and experimental investigations. This innovative approach offers a fresh perspective on enhancing the interfacial properties of MWCNT-reinforced composites by ensuring a more effective dispersion and interaction of these nanofillers within the adjacent matrix layers, thereby potentially leading to significant advancements in the field.

## 2. Materials and Methods

### 2.1. Materials

Poly(lactic acid) (PLA) is a promising polymer owing to its distinctive biocompatibility and desirable mechanical properties, as referenced by prior studies [24,39,40]. Its utilization in Fused Filament Fabrication (FFF) not only addresses concerns related to environmental pollution, but also leads to the creation of eco-friendly nanocomposites capable of biodegradation into harmless byproducts under standard degradation conditions [41,42]. Meanwhile, multi-walled carbon nanotubes (MWCNTs) are recognized for their exceptional potential in enhancing interlaminar properties, attributed to their high strength modulus and the advantageous characteristics of being high-aspect-ratio fillers [43,44]. This study involved PLA sourced from Shenzhen Creality 3D Technology Co., Ltd. (Shenzhen, China), demonstrating an average tensile strength of 57.8 MPa and an elastic modulus of 3.15 GPa. The MWCNTs, acquired from Suzhou Tanfeng Graphite Technology Co., Ltd. (Suzhou, China), were produced via chemical vapor deposition, boasting a length ranging from 3 to 12 μm and an average external diameter of 8–15 nm. The material’s purity surpassed 99 wt.%. The unique properties of PLA in combination with the reinforcing potential of MWCNTs were investigated in this study to understand their synergistic effects on the mechanical and structural attributes of the resulting nanocomposite materials.

### 2.2. Experiment Design

In contrast to conventional manufacturing techniques [29,30,31,32], the MWCNTs/PLA in-loop Hybrid Manufacturing (HM) process involves a unique method of combining plasma-actuated MWCNTs and PLA deposition via the Fused Filament Fabrication (FFF) technique [45]. This innovative process focuses on empowering MWCNTs through plasma activation during fabrication, enabling the creation of a stress transfer bridge formed by MWCNTs that permeates the entire interlayer interface. This bridge, facilitated by the plasma activation, enhances the interlayer connectivity by leveraging the nanoscale “reinforcing rib” effect of MWCNTs, thereby significantly improving the interlayer adhesion and anti-separation capabilities. To ensure the homogeneous dispersion of MWCNTs on the surface of the deposited PLA, a direct high-voltage current was employed between two electrodes to facilitate the rapid dispersion of MWCNTs in vaporized form. The method involved blending MWCNTs with deionized water in a ratio of 1:10, subjecting the mixture to ultrasonic vibration, and transporting it through a pipeline to the dispersion equipment’s head, where it was compacted into an electrode functioning as the cathode. Eventually, under plasma actuation, the deionized water within the cathode mixture underwent rapid vaporization, creating a localized pressure field that adhered the mixture effectively onto the substrate. The overall process of the in-loop HM fabrication is visually depicted in Figure 1, illustrating the sequential steps involved in this innovative manufacturing approach.

The PLA filament was deposited on the print bed as the first step. Then, the MWCNTs were sprayed uniformly on the surface of the deposited matrix using plasma-actuating dispersion equipment (as seen in Figure 1). The FFF deposition and plasma actuation were carried out independently and sequentially. In this research, the FFF process parameters’ effects on the interfacial strength were ignored due to the dominant strengthening effects of adding MWCNTs. The numbers of the in-loop switch between the dispersion and the deposition will become an HM process variable to control the MWCNT content in the ultimate print.

The ASTM D5023 standard was not employed in the present research due to its limitations in assessing interfacial properties, specifically, the failure observed from internal cracks in the three-point bending and trouser tear tests [46]. Consequently, it rendered the measurement of interfacial shear strength (IFSS) using the ASTM specimen unfeasible. As a result, this study introduced a specialized test specimen (depicted in Figure 2a) meticulously crafted to investigate the impact of Multi-Walled Carbon Nanotubes (MWCNTs) on the interfacial characteristics between adjacent beads. The specimen designed in this study comprised a 75 mm × 10 mm × 20 mm component with a 25 mm × 10 mm × 10 mm hole at one end and a 70 mm × 10 mm × 10 mm component, and the hole on the 75 mm × 10 mm × 20 mm component was symmetrical in the height direction. The 70 mm × 10 mm × 10 mm component can be embedded exactly into the aforementioned hole, and the length of the overlapping component between the two was 5 mm, as illustrated in Figure 2a.

To evaluate the effectiveness of the Hybrid Manufacturing (HM) method proposed, conventional MWCNTs/PLA composites were generated to establish a benchmark for comparison. The MWCNTs/PLA feedstock was prepared by blending MWCNTs with PLA pellets and subsequently extruding them using a single screw (1.75 mm) extruder at a temperature of 200 °C, with an extrusion speed of 45 cm/min, as per established methodologies [27,28,29,30,31]. The MWCNT content in the filament was regulated at 1 wt.% to prevent agglomeration issues, adhering to prior research findings [29,30,31]. The processing parameters employed for depositing the MWCNTs/PLA composite filament are detailed in Table 2, delineating the specific variables involved in this manufacturing process. 

The IFSS measurements were performed on an AI-7000-MU1 universal testing machine with a 10 KN load cell, and a crosshead rate of 5 mm/min, consistent with ISO 527:1997 [47], was adopted. All printed specimens, including the MWCNTs/PLA composites filament and three different MWCNT content specimens, were fabricated using the in-loop HM fabrication method by varying the number of switches between the processes. So that the varied MWCNT contents’ effects on the IFSS of MWCNTs/PLA composites could also be analyzed, each specimen was printed and tested in replications of five according to the aforementioned method. The IFSS results of different conditions were taken as the average of the measurements for the five repeated specimens.

The ultimate IFSS of the specimens was calculated using the following formula:(1)$$\tau_{IFSS}=\frac{F}{l\times w}$$
where F refers to the tested interfacial shear force and l and w represent the length and width of the contract area in the model, respectively.

## 3. Results and Discussion

### 3.1. The IFSS Test Results

Based on the test outcomes, the application of three layers of MWCNTs via spraying substantially amplified the Interfacial Shear Strength (IFSS) of the specimens by 30.50%, reaching 43.26 MPa, in comparison to those without any MWCNT addition. In stark contrast, the IFSS of the MWCNTs/PLA composites produced conventionally stood at a mere 9.06 MPa, representing only 27.3% of the pure PLA specimens and a meager 20.8% of the specimens treated with three layers of sprayed MWCNTs. These results underscore the pronounced significance of the proposed in-loop Hybrid Manufacturing (HM) method in enhancing interfacial properties. In the devised fabrication process, high-temperature-carrying MWCNTs were swiftly expelled from the interior of the plasma emitter under considerable pressure and made contact with the freshly deposited PLA matrix in various orientations. Subsequently, these non-parallel MWCNTs fused with the PLA material to form a structural bridge between the adjacent PLA layers, thus establishing a robust stress-transfer connection between the MWCNTs and PLA. This process notably improved the physical interlinking between the MWCNTs and PLA, effectively bridging the neighboring beads to prevent interface debonding, as visually represented in Figure 3. The orchestrated bonding and bridging mechanism of the MWCNTs within the printed PLA layers substantially contributed to the heightened strength and integrity of the interlayer interfaces, validating the efficacy of the proposed in-loop HM method in reinforcing the overall composite structure. 

### 3.2. The Effect of the Proposed HM Method on Activity and Conformation of PLA Molecule

MWCNTs/PLA nanocomposite filaments have traditionally exhibited significant tensile strength advantages, as documented in existing literature [28,29,30,31]. Typically, at a 1 wt.% concentration of MWCNTs, the tensile strength is reported to be around 1.34 times that of specimens made solely with pure PLA. However, the experimental findings in this study revealed a discrepancy: despite the heightened tensile strength, the interface properties of the product produced by MWCNTs/PLA nanocomposite filaments were notably poor. This inconsistency raises questions about the nature of the interface vis-a-vis the robust tensile strength observed. During the Fused Filament Fabrication (FFF) deposition process, the MWCNTs/PLA feedstock undergoes heating and extrusion through the nozzle. The alignment function of the inner nozzle wall results in the directional orientation of polymer chains and MWCNTs in parallel to both the extrusion direction of the filament and the printing platform. This orientation effect can constrain the interdiffusion of polymeric chains between filaments and hinder the mobility of extended polymer chains in newly printed adjacent filaments, primarily due to the alignment of MWCNTs parallel to the filaments. Consequently, this setup could lead to a reduction in the Interfacial Shear Strength (IFSS) of the specimens crafted from MWCNTs/PLA composite filaments.

The challenge lies in obtaining the desired conformation of polymer molecular chains and MWCNTs within the nozzle during the molding process. Molecular Dynamics (MD) simulation, a powerful tool referenced in the scientific literature [48,49,50,51], serves to effectively explore the morphological evolution of PLA chains and MWCNTs during the extrusion process. This simulation methodology operates at a molecular scale, shedding light on the underlying mechanisms that influence the interfacial properties of MWCNTs. It offers a comprehensive understanding of how the molecular configurations and interactions impact the overall behavior and properties of the MWCNTs within the composite, thereby unveiling insights from a molecular perspective.

Two 80 × 80 × 120 Å^3^ simulation models (PLA matrix extrusion model and MWCNTs/PLA composites filament extrusion model) were established in the commercial software Materials Studio 8.0 [49,51]. PLA chains with a polymerization degree of 15 were combined with double-walled carbon nanotubes (DWCNTs) to construct the PLA matrix with a density of 1.25 g/cm^3^, as well as MWCNTs/PLA composites. The PLA matrix and MWCNTs/PLA composites were extruded from the nozzle by a piston, forming a graphene-like structure with a 26.67 Å × 27.5 Å aperture (as illustrated in Figure 4). Upon the establishment of the simulated model, a geometric minimization step of 10,000 iterations (∆t = 1 fs) was initiated to achieve structural stability. Subsequently, dynamic equilibrium simulations were conducted under a constant volume, temperature, and particle number (NVT ensemble) to release residual stress and attain low-energy conformations. Following the equilibrium simulation, the models underwent an extrusion simulation representing the PLA matrix and MWCNTs/PLA composites, respectively. For simulating the extrusion process at 473 K, the nozzle remained fixed, while the piston was incrementally displaced along the *z*-direction with a displacement of 0.5 Å for each step of extrusion. The total displacement of the piston summed up to 45 Å. After each extrusion step, a dynamic simulation lasting 10 picoseconds (ps) with a time-step of 1 fs was performed to equilibrate the structure. An output was generated at every 1000 steps to capture frames for the subsequent calculations and analysis. This methodology enabled the investigation of the structural evolution and intermolecular interactions of the PLA matrix and MWCNTs/PLA composites during the extrusion process at a molecular level.

In Figure 4c,d, the impact of MWCNTs in the traditional manufacturing method is apparent in the observed restriction of the mean square displacement (MSD) and diffusion coefficient of the PLA matrix during the extrusion process. This limitation signifies a reduction in the mobility and activity of the matrix, thereby leading to an increased differentiation of interlayer molecular chains. However, in the proposed in-loop Hybrid Manufacturing (HM) approach, the MWCNTs do not undergo extrusion from the nozzle, escaping the orientation effects associated with nozzle-based extrusion. Furthermore, the plasma actuation provides high energy to the MWCNTs, amplifying the activity and radius of gyration (Rg) of the PLA molecular chains on the matrix’s surface. This elevated energy state promotes an enhanced interlayer diffusion of PLA molecules. To delve deeper, two distinct models sized 50 Å × 50 Å × 40 Å were constructed. In one simulation model, half of the MWCNTs content was vertically embedded within the PLA matrix, while the other identical model lacked any MWCNTs for comparative analysis. The PLA chains in both models corresponded to those in the extrusion model. Both systems underwent a 10,000 iteration step (∆t = 1 fs) geometry minimization to eradicate structural residual stress and achieve stability. Subsequently, dynamic equilibrium simulations were conducted under the NVT ensemble to release any remaining stress and attain low-energy conformations, adhering to established methodologies [42,48]. Following these steps, the two equilibrium systems were subjected to a temperature increase from 360 K to 500 K at a rate of 20 K every 30 picoseconds (ps) under constant NVT conditions. Thirty snapshots were stored during this process to calculate the MSD and Rg values of the PLA chains, as visually represented in Figure 5. These simulations allowed for a detailed assessment of the molecular dynamics and behavior of the PLA chains in the presence and absence of vertically embedded MWCNTs, shedding light on the interlayer diffusion dynamics within the PLA matrix.

### 3.3. The Effect of Introducing MWCNTs on Interface Energy 

In the proposed HM method, the plasma-actuated MWCNTs serve a crucial role in establishing a “stress transfer bridge” of carbon nanotubes between adjacent layers of the matrix. This bridge is formed due to the robust interfacial bonding and bridging effects between the MWCNTs and the PLA matrix chains. To comprehend and elucidate the intricate interactions between the PLA chains and MWCNTs, Molecular Dynamics (MD) simulations were employed to simulate the pullout process, revealing the underlying mechanisms by which the MWCNTs influenced the interfacial properties of the MWCNTs/PLA composites.

In the MD simulation, models representing the PLA separation and MWCNTs pullout from the MWCNT/PLA composites were constructed. These models integrated PLA chains with a polymerization degree of 15 and double-walled carbon nanotubes within the simulation environment. Following standard pre-processing procedures, equilibrium simulations were executed to attain stable configurations for subsequent analyses, mirroring established methodologies. Upon the completion of the equilibrium simulations, the models underwent the pullout simulation, as described in literature references [48]. The MWCNTs pullout process was conducted at 300 K, with the upper atoms of the MWCNTs being incrementally displaced along the *z*-direction at 5 Å for each pullout step. Similarly, the PLA in the separation part was moved along the *y*-direction with an identical displacement. After each pullout step, a dynamics simulation lasting 10 picoseconds (ps) with a time step of 1 fs was carried out to equilibrate the structure. Subsequently, output frames were generated at every 1000 steps to calculate the interfacial energy, offering insights into the interaction dynamics between the PLA chains and MWCNTs during the pullout process. These simulations provide a microscopic perspective on the behavior and characteristics of the MWCNTs and PLA interfaces, unveiling the mechanisms underlying their interfacial interactions. 

The interaction energy between MWCNTs and matrix PLA is calculated by Equation (1) below [49],
(2)$$\Delta E=E_{total}-\left(E_{MWCNTs}+E_{PLA}\right)$$
where $E_{total}$ refers to the total energy of the entire system, $E_{MWCNTs}$ refers to the energy of the MWCNTs, and $E_{PLA}$ refers to the energy of PLA.

The interfacial binding energy density is calculated by Equation (2) as follows [49],
(3)$$E_{\rho }=\frac{\Delta E}{A}$$

The interaction energy between the MWCNTs and the matrix PLA is calculated by Equation (2). A is the contact area between MWCNTs and the polymer matrix.

As depicted in Figure 6, during the extraction process of the carbon nanotubes, it was observed that the MWCNTs exhibited an energy absorption capacity. This characteristic contributes to delaying the propagation of interlayer cracks, thereby amplifying the nanoscale “reinforcing rib” effect of the carbon nanotubes. Consequently, this mechanism significantly augments the resistance to separation within the interlayer interface, ultimately leading to the enhancement of the interfacial properties of MWCNTs/PLA composites. This observation indicates that the MWCNTs play a critical role in fortifying the interlayer connections within the composite structure, which, in turn, strengthens the overall mechanical and structural integrity of MWCNTs/PLA composite materials.

## 4. Conclusions

This study introduced an innovative in-loop HM method that integrates plasma actuation and FFF technology to produce robust MWCNTs/PLA composites with an elevated interfacial strength. The impact of varying MWCNT content on the interfacial properties of the MWCNTs/PLA composites was thoroughly examined through Molecular Dynamics (MD) simulations and confirmed via experimental validation. The results distinctly indicated a notable influence of MWCNT content on the interfacial strength, revealing that a higher MWCNT content correlated with stronger interfacial properties. This innovative in-loop HM method holds promise for fabricating MWCNTs/PLA composites exhibiting enhanced interfacial properties suitable for diverse applications.

The proposed in-loop HM method efficiently mitigated the entanglement of PLA chains, enhancing the Interfacial Shear Strength (IFSS) of the specimens. Specifically, the qualitative and quantitative intuitive conclusions can be made:An evaluation of varying MWCNT contents’ impact on interfacial properties via MD simulations and experimental validation, revealing that a higher MWCNT content correlated with stronger interfacial properties.The in-loop HM method showcased a significant enhancement in the IFSS of specimens by 30.50%, reaching 43.26 MPa, 4.77 times higher than that of traditional MWCNTs/PLA composites.MD simulations suggest nozzle-based extrusion might impede PLA chain extension and diffusion, supporting the IFSS results observed in specimens made via the traditional method.

The incorporation of MWCNTs in the in-loop HM method also bolstered the mechanical properties of the PLA matrix. The high aspect ratio of the MWCNT content, coupled with the robust interfacial bonding energy between the MWCNTs and PLA molecules, notably bolstered the stiffness and strength of the PLA matrix. Additionally, the intermolecular interactions facilitated the vertical extension of the PLA chains along the MWCNTs, boosting their activity and effective interaction with the MWCNTs. This heightened activity strengthened the bond between the materials, ensuring an improved performance and stability. When a new layer of PLA material was deposited, the molecules infiltrated the MWCNT content, allowing for cross-linking with the existing PLA on the MWCNT surface. This robust interaction between the MWCNTs and PLA molecules further amplified the interfacial bonding energy, fortifying the PLA matrix and augmenting its strength and durability.

## Figures and Tables

**Figure 1 polymers-15-04426-f001:**
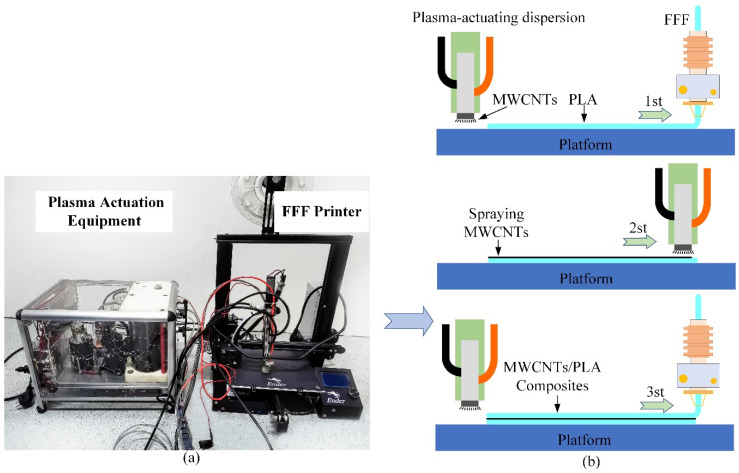
(**a**) The hybrid fabrication equipment: MWCNTs’ dispersion equipment and FFF printer; and (**b**) the schematic of hybrid fabrication of MWCNTs/PLA composites [45].

**Figure 2 polymers-15-04426-f002:**
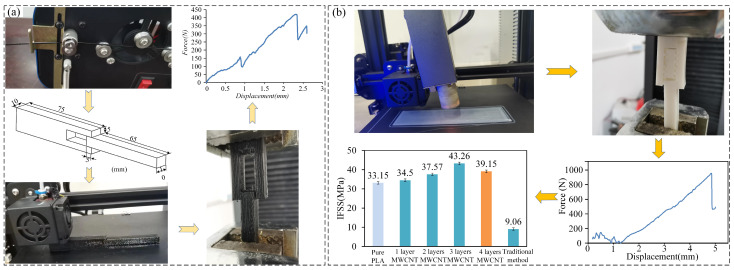
(**a**) The IFSS test of the specimen made by MWCNTs/PLA composites filament. (**b**) IFSS test of three different MWCNTs contents by the proposed in-loop HM process.

**Figure 3 polymers-15-04426-f003:**
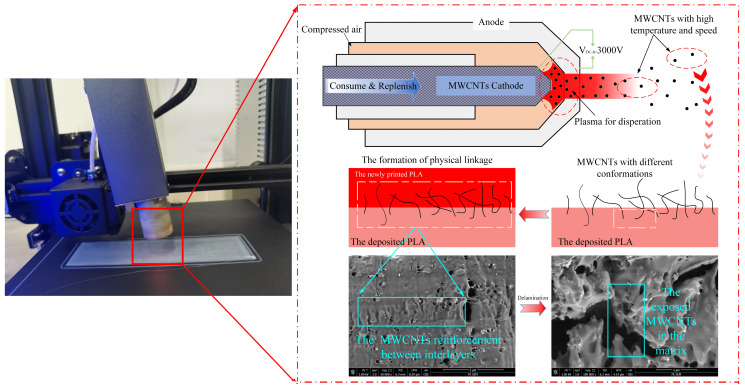
The formation of the physical interlinkage between the adjacent PLA layers thanks to MWCNTs.

**Figure 4 polymers-15-04426-f004:**
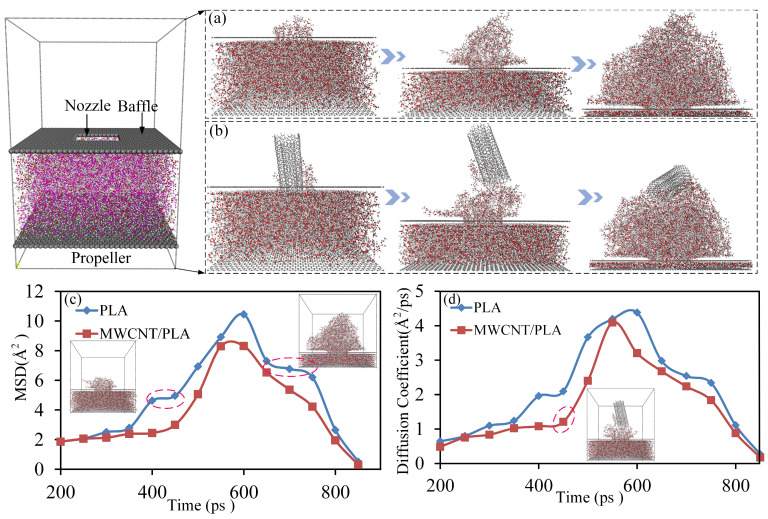
The extrusion simulations of PLA matrix and MWCNTs/PLA composites from the nozzle by MD. (**a**) The extrusion process of PLA matrix; (**b**) the extrusion process of MWCNTs/PLA composites; (**c**) the mean square displacement of PLA chains in the two different processes; and (**d**) the diffusion coefficient of PLA chains in the two different extrusion processes.

**Figure 5 polymers-15-04426-f005:**
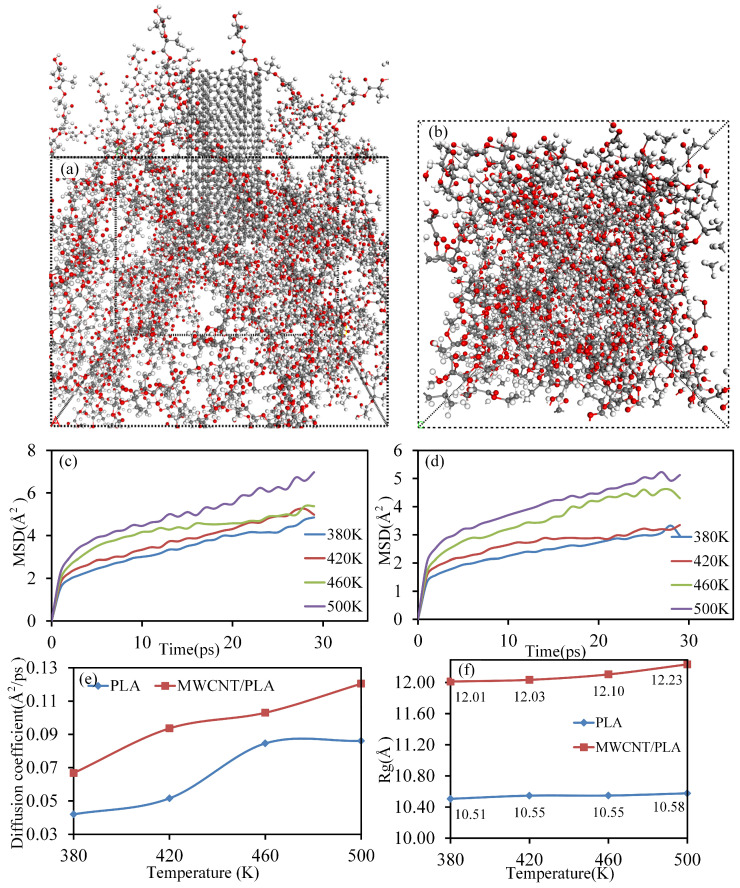
The change calculations of PLA molecular activity and conformation with temperature. (**a**) The MWCNTs/PLA heating simulation model; (**b**) the PLA heating simulation model; (**c**) the MSD values of the PLA chains in the MWCNTs/PLA composites with an increase in temperature; (**d**) the MSD values of the PLA chains in the pure PLA matrix with an increase in temperature; (**e**) the diffusion coefficients of the PLA chains in the two models with the increase in temperature; and (**f**) the Rg values of the PLA chains in the two models with the increase in temperature.

**Figure 6 polymers-15-04426-f006:**
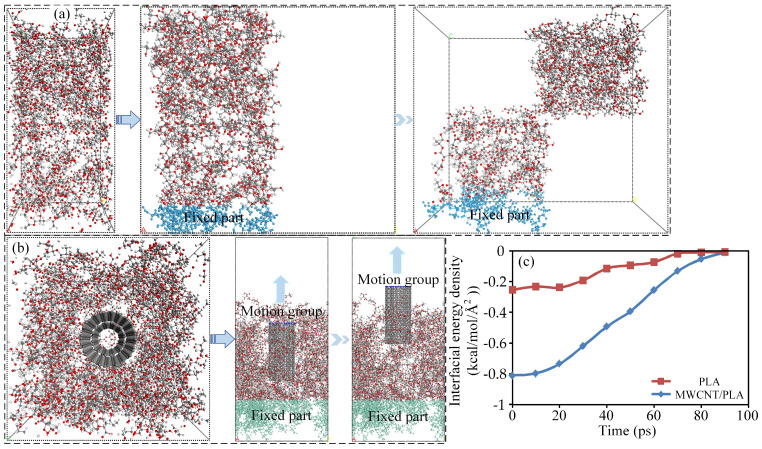
(**a**) The separation process of the adjacent PLA layers; (**b**) the pullout process of MWCNTs content from MWCNTs/PLA composites by in-loop HM; and (**c**) the interfacial energy density between MWCNTs content, PLA matrix, and the separated PLA layers.

**Table 1 polymers-15-04426-t001:** The current strategies for enhancing interfacial shear strength.

Study	Method	Key Findings/Technical Data
Caminero et al. [5] Ning et al. [18]	Carbon Fiber-Reinforced	—Tensile strength increased by 20–40% —Flexural strength increased by 35% —Enhanced stiffness and durability
Sweeney et al. [19] Hart et al. [20]	Heat-Treating PLA (FFF)	—Tensile strength increased by 10–15% —Layer adhesion strength improved due to crystalline structure —Enhanced overall mechanical properties, especially stiffness
Alireza et al. [35] Li et al. [36]	Ultrasonic vibration	—Interlayer adhesion increased by 10% —Tensile strength increased by 10.7%
Sun et al. [37] Wu et al. [38]	MWCNTs-Reinforced	—Inter-laminar shear strength increased by 22.7% —The mechanical properties and electromagnetic shielding efficiency were improved

**Table 2 polymers-15-04426-t002:** Printing parameters for the FFF printed specimens.

Parameter	Value
Nozzle diameter	0.4 mm
Layer thickness	0.2 mm
Top/bottom solid layers	0.8 mm
Infill speed	80 mm/s
Printing speed (shells)	25 mm/s
Infill density	100%
Extruder temperature	200 °C
Bed temperature	60 °C
Outline perimeters/shells	2
Raster angle	±45°

## Data Availability

The data presented in this study are available on request from the corresponding author.

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
