# Peer review of "The Interface Strengthening of Multi-Walled Carbon Nanotubes/Polylactic Acid Composites via the In-Loop Hybrid Manufacturing Method"

_polymers, 2023, doi:10.3390/polym15224426_

Round 1

Reviewer 1 Report

Comments and Suggestions for Authors

The paper shows an interesting approach to the use of additional postprocessing after layer distribution during material extrusion technology.  The introduction is very well written, the authors described an issue with the adhesive connection between layers and supported it with proper literature. Such an approach allowed to formation of a reasonable scientific hypothesis. Throughout the whole manuscript, I found the following issues: 

1. Please use proper nomenclature related to ASTM/ISO 52900 standard regarding nomenclature (MEX is the leading group where FFF technology is placed). 

2. I cannot understand material selection - PLA has a quite high UTS but it has a very small VICAT/HDT temperature and low UV resistance which is why it is not proper to use in industrial applications. Please better justify your selection. 

3. Provide more data about the increase in the estimated printing time - how your technology would affect the total printing time? 

4. Provide input data that you put into the software to create your models. 

5. I suggest putting your conclusions by showing the main outcomes in specific points. 

Author Response

Response to Reviewer 1:

Dear reviewer,

Thank you so much for your comments.

Here are our responses to your related concerns.

  1. Please use proper nomenclature related to ASTM/ISO 52900 standard regarding nomenclature (MEX is the leading group where FFF technology is placed). 

Response: We understand that MEX is listed as one of the 7 AM processes by ISO. However, in the case of “MEX”, we have FFF. But we also have paste extrusion. And concrete extrusion. These are utterly different processes yet are all “material extrusion”. In this paper, we only focus on the fused filament material extrusion, that is why we want to use FFF in the article.

  1. I cannot understand material selection - PLA has a quite high UTS but it has a very small VICAT/HDT temperature and low UV resistance which is why it is not proper to use in industrial applications. Please better justify your selection. 

Response: Studying PLA (Polylactic Acid) remains relevant and valuable for several reasons, despite its limitations in certain industrial applications. Given that PLA typically exhibits minimal warping in parts and considering our ongoing development of a new hybrid process, we aim to minimize part distortion to ensure that it does not compromise the accuracy of the experimentally obtained data. It's worth noting that PLA is known for its non-toxic and safe properties.

  1. Provide more data about the increase in the estimated printing time - how your technology would affect the total printing time? 

Response: The total printing time remains unaffected by our hybrid method because it operates seamlessly without any interruptions during the material deposition process. Nanotubes are dispensed through a deposition head positioned adjacent to the FFF nozzle. As soon as the FFF nozzle completes one layer, the nanotubes are immediately sprayed onto that layer. This entire process is completed in a matter of milliseconds.

  1. Provide input data that you put into the software to create your models. 

Response:  The specimen designed in this study comprised a 75 mm × 10 mm × 20 mm component with a 25 mm × 10 mm × 10 mm hole at one end and a 70 mm × 10 mm × 10 mm component, and the hole on the 75 mm × 10 mm × 20 mm component was symmetrical in the height direction. The 70 mm × 10 mm × 10 mm component can be embedded exactly into the aforementioned hole, and the length of the overlapping component between the two was 5 mm, as illustrated in Fig. 2a.

  1. I suggest putting your conclusions by showing the main outcomes in specific points. 

Response: We have revised the whole manuscript, including the conclusion section. Specifically, we used bullet points to highlight our main discovery.

Reviewer 2 Report

Comments and Suggestions for Authors

Authors should add more results of compatibility of PLA with MWCNTs and their physical properties.

Comments on the Quality of English Language

The manuscript should be improved with native speaker.

Author Response

Response to Reviewer 2:

Dear reviewer,

Thank you so much for your comments.

Here are our responses to your related concerns.

1.Authors should add more results of compatibility of PLA with MWCNTs and their physical properties. 

Response: According to our previous study, the surface of MWCNTs is inert and there are no functional groups that can react with PLA, and the interaction between the two is composed of van der Waals force and weak electrostatic force. The interface, as a weak point in FFF part, is a vulnerable point to failure under load. This article utilizes the nanoscale enhancement effect of MWCNTs and plasma-actuation dispersion technology to spray MWCNTs between adjacent matrix beads, exerting the nanoscale reinforcement of MWCNTs and improving the interface bonding strength of the matrix. The effect of this method is shown in Fig. 2b.

  1. The manuscript should be improved with native speaker.

Response: The full text has been checked and the language of the article has been revised and edited.

Reviewer 3 Report

Comments and Suggestions for Authors

This work reported a new in-loop hybrid manufacturing method to fabricate   carbon nanotube (MWCNTs)/polylactic acid (PLA) hybrid composites. Overall, the study is soild and well presented. There are still a few points need to be improved for publication. 

1. It would be great to add a table to compare with other materials for interfacial shear strength to show the effectiveness of the hybrid materials. 

2. How do authors optimize the parameters showing in table 1 such as printing speed?

3. Why choose PLA? More information can be provided in the introduction. Is that possible to change PLA to other printing materials for the same purpose?

Comments on the Quality of English Language

Overall, english could be polished.

Author Response

Response to Reviewer 3:

Dear reviewer,

Thank you so much for your comments.

Here are our responses to your related concerns.

  1. It would be great to add a table to compare with other materials for interfacial shear strength to show the effectiveness of the hybrid materials.

Response: We have added a table to the manuscript in the introduction section.

Study

Method

Key Findings/Technical Data

Caminero et al. [5]

 Ning et al. [18]

Carbon Fiber-Reinforced

- Tensile strength increased by 20-40%

- Flexural strength increased by 35%

- Enhanced stiffness and durability

Sweeney et al. [19]

Hart et al. [20]

Heat-Treating PLA (FFF)

- Tensile strength increased by 10-15%

-Layer adhesion strength improved due to crystalline structure

-Enhanced overall mechanical properties, especially stiffness

 Alireza et al. [38]

 Li et al. [39]

Ultrasonic vibration

- Interlayer adhesion increased by 10%

-Tensile strength increased by 10.7%

Sun et al. [40]

Wu et al. [41]

MWCNTs-Reinforced

-Inter-laminar shear strength increased by 22.7%

-The mechanical properties and electromagnetic shielding efficiency were improved

  1. How do authors optimize the parameters showing in table 1 such as printing speed?

Response: We have published research papers that focus on optimizing process planning to enhance tensile strength in both Fused Filament Fabrication (FFF) and Laser Powder Bed Fusion (LPBF) processes. You can find the references for these papers below.

Building upon the models we developed earlier and drawing from existing literature, we are utilizing the parameters listed in Table 1. These parameters are selected to ensure the feasibility of the process while aiming to achieve specific tensile strength targets based on our parameter choices.

  1. Why choose PLA? More information can be provided in the introduction. Is that possible to change PLA to other printing materials for the same purpose?

Response: Studying PLA (Polylactic Acid) remains relevant and valuable for several reasons, despite its limitations in certain industrial applications. Given that PLA typically exhibits minimal warping in parts and considering our ongoing development of a new hybrid process, we aim to minimize part distortion to ensure that it does not compromise the accuracy of the experimentally obtained data. It's worth noting that PLA is known for its non-toxic and safe properties.

And certainly, we can use other materials other than PLA for the base material in the proposed hybrid process. We hypothesize to see certain strength improvement if we choose PETG or ABS material, but the specific amount increase needs to be corrected based on some preliminary tests.

Round 2

Reviewer 1 Report

Comments and Suggestions for Authors

The authors made some improvements, but still, there are some parts that need to be corrected:

1. It is not prohibited to use a commercial type of technology, but before using it you should introduce MEX technology first. 

2. The explanation completely misses the issue mentioned in the original comment. "(...) PLA is not proper to use in industrial applications. Please better justify your selection." Warping is a typical, technological issue that affects dimensional accuracy. To be more specific - point out the industrial field of your solution... 

Author Response

Dear reviewer,

Thank you so much for your comments.

Here are our responses to your related concerns.

  1. It is not prohibited to use a commercial type of technology, but before using it you should introduce MEX technology first. 

Response: We have added the introduction of MEX in the paper, as following:

Among the various AM methods, Material Extrusion (MEX) stands as a prominent choice. MEX leverages a swift material deposition process, achieved through selectively dispensing of polymers through a nozzle.

  1. The explanation completely misses the issue mentioned in the original comment. "(...) PLA is not proper to use in industrial applications. Please better justify your selection." Warping is a typical, technological issue that affects dimensional accuracy. To be more specific - point out the industrial field of your solution... 

Response: The performance of PLA (Polylactic Acid) still meets the requirements of many industries, despite its limitations in certain industrial applications. One the one hand, PLA is a promising alternative to traditional petrochemical polymers, with distinct advantages of biocompatibility and desired mechanical properties. Moreover, as one of the most widely used biopolymer materials in FFF, it addresses concerns regarding environmental pollution and forms eco-friendly nanocomposites that can biodegrade into safe byproducts under standard degradation conditions. On the other hand, PLA is used as a drug transport material, tissue engineering scaffold material and bone repair material in biomedical field, and is widely used as an accessory in the automotive industry.
